# Process Monitoring of Quality-Related Variables in Wastewater Treatment Using Kalman-Elman Neural Network-Based Soft-Sensor Modeling

**Yiqi Liu * , Longhua Yuan, Dong Li, Yan Li and Daoping Huang**

School of Automation Science and Engineering, South China University of Technology, Guangzhou 510640, China; 201920116203@mail.scut.edu.cn (L.Y.); lddscut@163.com (D.L.); liyan@scut.edu.cn (Y.L.); audhuang@scut.edu.cn (D.H.)
* Correspondence: aulyq@scut.edu.cn; Tel./Fax: +86-20-87114189

**Abstract:** Proper monitoring of quality-related but hard-to-measure effluent variables in wastewater plants is imperative. Soft sensors, such as dynamic neural network, are widely used to predict and monitor these variables and then to optimize plant operations. However, the traditional training methods of dynamic neural network may lead to poor local optima and low learning rates, resulting in inaccurate estimations of parameters and deviation of predictions. This study introduces a general Kalman-Elman method to monitor the effluent qualities, such as biochemical oxygen demand (BOD), chemical oxygen demand (COD), and total nitrogen (TN). The method couples an Elman neural network with the square-root unscented Kalman filter (SR-UKF) to build a soft-sensor model. In the proposed methodology, adaptive noise estimation and weight constraining are introduced to estimate the unknown noise and constrain the parameter values. The main merits of the proposed approach include the following: First, improving the mapping accuracy of the model and overcoming the underprediction phenomena in data-driven process monitoring; second, implementing the parameter constraint and avoid large weight values; and finally, providing a new way to update the parameters online. The proposed method is verified from a dataset of the University of California database (UCI database). The obtained results show that the proposed soft-sensor model achieved better prediction performance with root mean square error (RMSE) being at least 50% better than the Elman network based on back propagation through the time algorithm (Elman-BPTT), Elman network based on momentum gradient descent algorithm (Elman-GDM), and Elman network based on Levenberg-Marquardt algorithm (Elman-LM). This method can give satisfying prediction of quality-related effluent variables with the largest correlation coefficient (R) for approximately 0.85 in output suspended solids (SS-S) and 0.95 in BOD and COD.

**Keywords:** soft-sensor; Kalman filter; Elman network; adaptive noise

## 1. Introduction

During recent decades, increased awareness about the negative impact of eutrophication on the quality of water bodies and advances in environmental technology have given rise to more stringent wastewater treatment requirements and regulations [1,2]. Currently, proper monitoring of wastewater plants is one of the main challenges for water utilities worldwide, with significant environmental and cost-saving implications. Measurement and monitoring of effluent qualities are one of most important aspects. Because of the extreme working conditions, the highly complex processes of microbial growth and large measurement delays, the measurement of effluent quality parameters, such as $BOD_5$ (biochemical oxygen demand for 5 days), COD (chemical oxygen demand), and TN (total nitrogen) is usually difficult [2,3]. To describe the physical, chemical, and biological reactions in wastewater, many differential equations are required, which discourages process model construction. Because prior knowledge is not required, data-driven soft-sensor technology

has become the most commonly-used method to measure the quality-related variables of biological treatment processes in the wastewater treatment. Essentially, data-driven soft-sensor technology aims to construct a certain mathematical model to describe the relationship between input and output variables, to predict hard-to-measure variables without necessarily resorting to an accurate mechanism model [4].

Generally, the physical, chemical, and biological phenomena associated with treatment units always lead to many difficult-to-measure quality-related variables, such as biochemical oxygen demand (BOD), COD, and TN, thus complicating the reliable management of WWTPs [5]. Even though some hardware sensors have been developed, the unacceptable costs and unreliability of the corresponding hardware sensors always make them inadequate for large amounts of WWTPs, particularly in rural areas or in developing countries. With the recent development of machine learning, data-driven soft sensor technology has been widely used in wastewater treatment processes [6] and has become an important component of advanced process control technology [7]. Specifically, neural network-based data-driven soft-sensors have become one of the most active research fields due to their strong nonlinear mapping ability, network topology, and robustness [8], as well as their independence from mathematical models [9]. However, traditional feedforward neural networks are difficult to apply for wastewater treatment processes due to failure to deal with strong dynamical issues. Wastewater processes always exhibit strong nonlinear dynamics (such as extreme weather conditions) and have coupling effects among the variables (such as recycling of the sludge in the secondary clarifier). These factors add more necessity to the wide applications of adaptive neural networks for soft-sensor modeling in WWTPs. Internal feedback connections between processing units were augmented into a neural network (NN) to formulate a recursive neural network (RNN) and to enhance the dynamic prediction ability. As one of the most typical RNN, the dynamic characteristics of Elman network has been proved by Guan et al. [10] and Liang [11].

Recently, many methods have been proposed to train RNNs [12], such as the real-time recurrent learning (RTRL) [13] algorithm, back propagation through time (BPTT) [14], momentum gradient descent algorithm, and Levenberg-Marquardt (LM) algorithm [15]. All of these methods have been widely used and exhibit superior capabilities. Unfortunately, they may be plagued by converging into poor local optima and a low learning rate [16,17]. The Kalman filter (KF) [18] provides an inherently recursive solution to the optimal filtering estimation problem and to reducing forecast uncertainty. Moreover, the KF can be implemented in sequential mode and does not require an inversion of the approximate Hessian matrix. Unfortunately, the KF is usually only tractable for linear systems. To apply the Kalman framework to nonlinear systems, the extended Kalman filter (EKF) [18] and unscented Kalman filter (UKF) [19] are two common usages of nonlinear Kalman filters. The EKF, an effective second-order algorithm, can be applied to estimate the weights of RNN. The use of the EKF for training neural networks has been developed by Singhal and Wu [20] and Puskorious and Feldkamp [21]. Actually, the first-order truncated Taylor series expansion employed by EKF can induce large estimation errors and lead to divergence of the filter itself. These can be addressed using UKF. The UKF consistently outperforms the EKF in terms of prediction and estimation error [22,23]. Rudolph van der Merwe et al. proposed the algorithm of numerically effective square root form of UKF (SR-UKF) [24], which can effectively preserve the symmetry and positive definiteness of the updated covariance. To date, the use of the group of UKF algorithms has been further expanded within the general field of probabilistic inference [25] and machine learning [26]. However, few researches devoted to process monitoring of effluent qualities in wastewater treatment. This paper adopts the SR-UKF to train the standard Elman network. Considering the uncertainty of statistical characteristics of the system noises in the actual system will affect the prediction performance [27], this paper introduces an adaptive Sage-Husa noise estimation method to solve this problem. In addition, it is important that there is no restriction on weight values in the traditional training methods for Elman network, which could lead

to inefficient standard Elman network model building. Therefore, this paper applies a parameter constraint algorithm to avoid large weight values.

In this study, an Elman network based on SR-UKF was proposed to monitor the effluent qualities, which can improve the prediction performance and ensure wide applicability of the Elman network. The contributions of this paper were mainly from three aspects. (i) First, this paper combined the Elman network with the Joseph form of UKF to build a soft-sensor model, thus being further able to enhance the accuracy and reliability of a soft-sensor. (ii) Second, this study proposed an adaptive method to estimate the system noise in real-time. This can ensure that the soft-sensor model is capable of achieving accurate prediction in case of suffering unknown disturbances. (iii) Finally, to ensure model weights are not updated aggressively and to guarantee that a robust model can be derived, a constraint algorithm was used to constrain the model parameter values to a reasonable range. In the proposed algorithm, the purpose of limiting weights to a specified range is mainly to ensure that the model can achieve smooth mapping, which is able to restrict the model complexity and avoid overfitting. Moreover, a weight constraining algorithm can guarantee the involvement of prior knowledge by setting up a proper weight range and then can meet the parameter setting requirements according to the specific applications. Instead of using weights randomly, limiting weight to a specified range ensures that the algorithm is manageable and that the reliability of a soft sensor can be further achieved.

Additionally, it is important to note the contributions of the proposed soft-sensors for process monitoring and management of WWTPs. The physical, chemical, and biological reactions of wastewater treatment processes often add significant nonlinearity and dynamics for modeling and result in the degradation of standard prediction models. Thus, to properly monitor quality-related but hard-to-measure effluent variables, this study uses of the proposed algorithm to update the weights of the Elman network and to improve the prediction performance. This will, in turn, provide a new way to prevent the degradation of predictive performance. Moreover, to decrease the monitoring costs, the proposed soft-sensor is further extended to a multioutput model, which can simultaneously monitor multiple effluent qualities simultaneously.

The rest of this article is organized as follows. In Section 2, basic materials and preliminary methods are given. Section 3 addresses the proposed models and validated case study materials. Section 4 provides detailed results and discussions for the case study. Finally, conclusions are made.

## 2. Preliminary Materials and Methods

In this section, the concepts and preliminary knowledge of the Kalman filter, SR-UKF, and Elman neural network are briefly reviewed.

### 2.1. Kalman Filter

Based on the state space equation and measurement equation, the Kalman filter is able to estimate the state of the process by finding the minimum mean square error. In particular, when the system suffers from serious uncertainties, it provides an effective way to estimate the state of the process, which can work even in the case of serious uncertainties. The discrete-time linear system is shown as follows:

$$\begin{cases} x_k = Ax_{k-1} + Bu_{k-1} + w_k \\ z_k = Hx_k + v_k \end{cases} \tag{1}$$

where $x_k$ and $z_k$ are the system state and the measurement vector at time step $k$, respectively. $A$ is the state transition matrix and $B$ represents the influence matrix of the exogenous input $u$ on the state $x$. $H$ is the observation matrix. $w_k$ and $v_k$ are process noise and measurement noise, respectively, and both of them are white Gaussian noise with $w_k \sim N(0, R)$, and $v_k \sim N(0, Q)$. $R$ and $Q$ are the process noise covariance and measurement noise covariance, respectively.

The Kalman filter algorithm mainly consists of two steps: state space update equations and measurement update equations. The state space update step uses the optimal state in the current time to estimate the next prior. In the measurement update step, the priori is updated with the innovation contained in the latest observation. For more detail, see [18].

The Kalman filter is usually only tractable for linear systems. To apply the Kalman framework to nonlinear systems, it is necessary to apply nonlinear Kalman filters, such as EKF and UKF. The EKF usually uses the first-order truncated Taylor series expansion to locally linearize the nonlinear function. However, Taylor series expansion often results in large errors in the estimated statistics of the state posterior distribution. Additionally, it also fails to deal with highly nonlinear system because the higher-order terms of the Taylor series expansion are always neglected. Different from the EKF, the UKF obtains high-order mean and covariance by approximating the probability density function through a set of important sample points, which can avoid the errors caused by the local linearity assumption and have higher estimated accuracy than the EKF.

### 2.2. SR-UKF Algorithm

As a nonlinear filtering method, the UKF algorithm is able to capture the posterior mean and covariance of state random variables by selecting a set of important samples (also called a set of sigma sample points) rather than explicitly approximating the nonlinear state space and observation model with all samples.

These sigma points can accurately capture the mean and covariance of any nonlinear function with the second order when the state variables are propagated through a nonlinear system. The model residual errors are summarized in the third and higher orders [25]. Thus, it can effectively solve the problem of local linear assumptions and improve the estimation accuracy and robustness of the standard Kalman filter.

For an *L*-dimensional random variable $x$ with the mean $\bar{x}$ and covariance $P_x$, the sample points $\chi$ and the corresponding weight $W$ were selected by the scaled unscented transformation (SUT) [28] method:

$$
\begin{cases}
\chi_0 = \bar{x} & i = 0 & w = \frac{\lambda}{L+\lambda} & i = 0 \\
\chi_i = \bar{x} + \left(\sqrt{(L+\lambda)P_x}\right)_i, & i = 1, \cdots, L & w_0^{(c)} = \frac{\lambda}{2(L+\lambda)} + \left(1 - \alpha^2 + \beta\right), & i = 0 \\
\chi_i = \bar{x} - \left(\sqrt{(L+\lambda)P_x}\right)_i, & i = L+1, \cdots, 2L & w_i^{(m)} = w_i^{(c)} = \frac{\lambda}{2(L+\lambda)}, & i = 1, \cdots, 2L
\end{cases}
\tag{2}
$$

where $\lambda = \alpha^2(L + \kappa) - L$ is the scale parameter, $\left(\sqrt{(L+\lambda)P_x}\right)_i$ represents the *i*th column of the square root of the matrix $(L + \lambda)P_x$, $\kappa$, and $\alpha$ and $\beta$ are optional parameters. The parameter $\kappa(\kappa \geq 0)$ is used to ensure the semipositive definition of the covariance matrix. The parameter $\alpha(0 \leq \alpha \leq 1)$ controls the "size" of the sigma points, which limits identification to a small number to avoid the influence of strong nonlinearity on nonlocal sampling. The parameter $\beta(\beta \geq 0)$ is a nonnegative weighted term, which can be used to retain the information of higher-order moments.

Even though UKF works well, two shortcomings of the UKF need to be addressed. First, in iterative learning, the generation of sigma points needs to calculate the square root of the matrix in Equation (2). Second, due to the random error and calculation accuracy, the positive definite nature and symmetry of the state error covariance may be broken down, resulting in divergence of the algorithm. To address these problems, this study makes full use of the numerically effective square root form of UKF, namely SR-UKF [24]. SR-UKF introduces three powerful linear algebra techniques: QR decomposition, Cholesky factor updating, and effective least squares. QR decomposition algorithm and Cholesky factor updating algorithm. These improvements can effectively avoid the ill square root operation of the error covariance and the negative effect of least squares (right division operator "/" in MATLAB), and then avoid the matrix inverse operation during the calculation.

Assuming that $P$ is the prediction error covariance matrix, according to:

$$
P = AA^T = R^TQ^TQR = R^TR = SS^T
\tag{3}
$$

The QR decomposition of the square root factor $A^T$ of $P$ can return the upper triangular matrix $R$, while $S = R^T$ is the lower triangular matrix, which can be propagated and updated by $S$ to avoid the square root operation in each iteration. To update $S$, the Cholesky factor updating was introduced. For a Cholesky factor $S$ of $P = AA^T$, its vector updates $\check{P} = P + \sqrt{v}uu^T$ can be written as:

$$\check{S} = cholupdate\{S, u, \pm v\} \tag{4}$$

If $u$ is a matrix rather than a vector, continuous updates can be made using each column of $u$. The Kalman gain can be formulated as:

$$K_k = P_{x_k z_k} P_{\tilde{z}_k \tilde{z}_k}{}^{-1} = \left(P_{x_k z_k} / S_{\tilde{z}_k}^T\right) / S_{\tilde{z}_k} \tag{5}$$

where the operator "/" in MATLAB can avoid the inversion of the error covariance matrix.

The abovementioned techniques can improve the numerical stability of the algorithm and maintain the positive definite nature and symmetry of the covariance matrix. For more information, please refer to [24].

### 2.3. Elman Neural Network

The Elman network, is a typical local regression network [29]. The Elman network is a recurrent neural network with local memory units and local feedback connections (Figure 1). The recurrent layer augments the states at the previous moment and the network input at the current moment as the input of the hidden layer. This internal state feedback can increase the dynamic characteristics of the network.

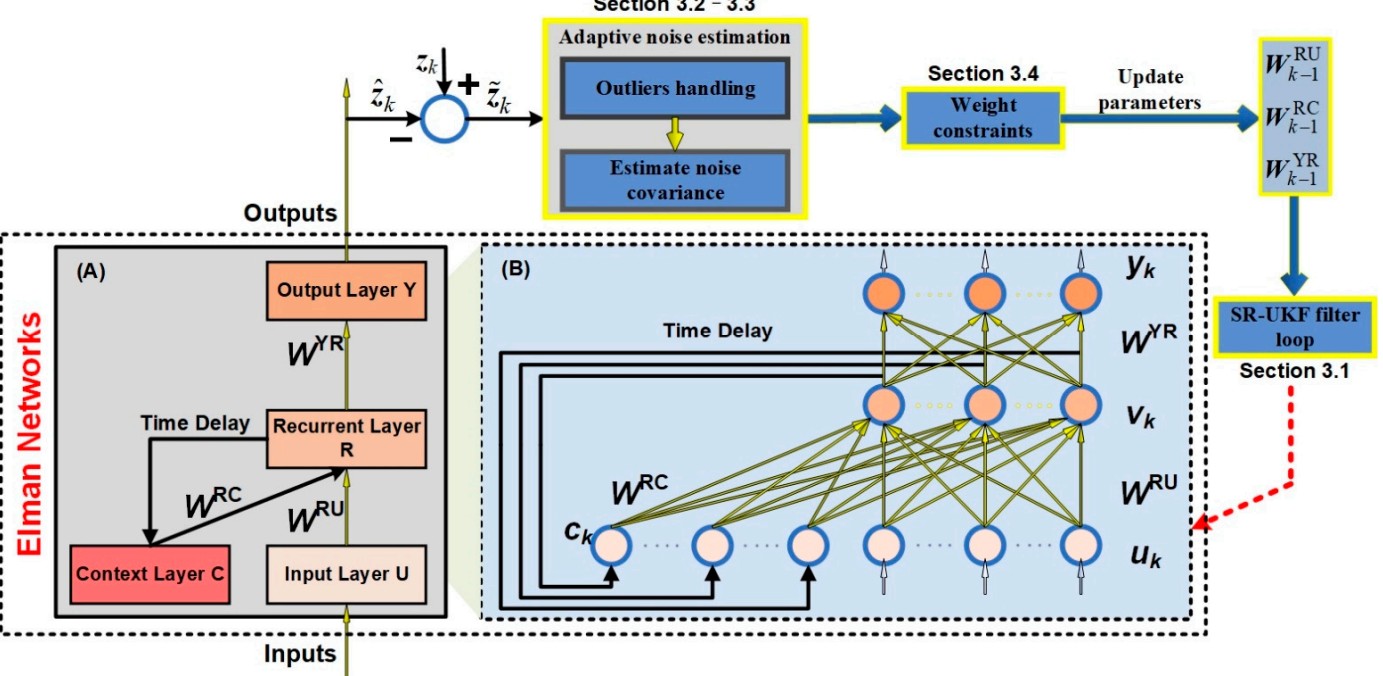

**Figure 1.** (**A**) The simplified topology of the Elman neural network; (**B**) A detailed expansion of module A.

The Elman network mainly consists of four layers: the input layer U, the hidden layer (or recurrent layer) R, the context layer C and the output layer Y. $W^{RU}$, $W^{RC}$, and $W^{YR}$ represent the connection weight matrix from the input layer to the hidden layer, the context layer to the hidden layer and the hidden layer to the output layer, respectively. The output of the hidden layer after unit delay is used as the input of the next context layer, $c_k = v_{k-1}$, and it is applied to the hidden layer through the matrix $W^{RC}$.

Given the input $\boldsymbol{u}_k \in \Re^{N_i}$ at time step $k$, the output of the Elman network can be calculated as follows:

$$\widetilde{\boldsymbol{v}}_k = \boldsymbol{W}^{\mathrm{RU}}\boldsymbol{u}_k + \boldsymbol{W}^{\mathrm{RC}}\boldsymbol{v}_{k-1} + \boldsymbol{b}_h \tag{6}$$

$$\boldsymbol{v}_k = f_h(\widetilde{\boldsymbol{v}}_k), \boldsymbol{v}_k \in \Re^{N_h} \tag{7}$$

$$y_k = f_o\left(\boldsymbol{W}^{\mathrm{YR}}\boldsymbol{v}_k + \boldsymbol{b}_o\right), y_k \in \Re^{N_o} \tag{8}$$

where $\boldsymbol{b}_h$ and $\boldsymbol{b}_o$ are the bias vectors of the hidden layer and output layer respectively, and $f_h(\cdot)$ and $f_o(\cdot)$ are the activation functions of the hidden layer and output layer, respectively. $N_i$, $N_h$, and $N_o$ represent the number of nodes in the input layer, hidden layer and output layer, respectively.

The RTRL, BPTT, momentum gradient descent algorithm, and LM algorithms are the most commonly used methods to train Elman neural network. In the following section, the abovementioned methods are compared with the SR-UKF algorithm in this study to verify the effectiveness.

## 3. Proposed Prediction Model and Validation Materials

In this section, the proposed SR-UKF algorithm in parameter estimation for the Elman neural network will be described. First, an overall view of the proposed SR-UKF algorithm is given in Section 3.1; The other subsections provide more details about the adaptive noise estimation, weight constraint and handling outliers, which are the important components of the proposed SR-UKF algorithm (Figure 1).

This figure shows an overall view of the proposed SR-UKF algorithm. The module "Outliers handling" controls the amplification of observation error covariance and will combine the module "Estimate noise covariance" to achieve the adaptive estimation the unknown noise. The module "Weight constraints" adds parameter constraints on weight (state) values. Actually, the module "Adaptive noise estimation" and the module "Weight constraints" are independent modules. Both of the "Adaptive noise estimation" and "Weight constraints" modules could be removed, which represent the constant noise and unconstrained weights, respectively. Module A is the simplified topology of the Elman network in module B. Module B gives a more detailed expansion of module A to increase the visibility of the proposed algorithm.

### 3.1. Elman Network Based on SR-UKF (Elman-SR-UKF)

The training process of traditional neural networks is to adjust the parameters (or weights). Generally, the neural network method based on the Kalman filter regards the learning process of the network as the dynamic parameter estimation of the nonlinear system. That is, the weight vector in Elman is taken as the state of the system, and the weights of the network are constantly updated with the time sequence by minimizing the mean square error between the target output and the estimated output, which can improve the training accuracy. The state space model of the neural network can be rewritten as:

$$\begin{cases} \boldsymbol{x}_{k+1} = \boldsymbol{x}_k + \boldsymbol{r}_k \\ \boldsymbol{z}_k = h(\boldsymbol{x}_k, \boldsymbol{u}_k) + \boldsymbol{q}_k \end{cases} \tag{9}$$

where $\boldsymbol{x}_k$ is the state vector composed of weight matrices and biases, $\boldsymbol{u}_k$ represents a given input, $\boldsymbol{z}_k$ is the model output, and $h(\cdot)$ is the neural network function. Moreover, $\boldsymbol{r}_k$ and $\boldsymbol{q}_k$ are the zero-mean Gaussian process noise vector and measurement noise vector with covariance $\boldsymbol{R}_k$ and $\boldsymbol{Q}_k$, respectively. The nonlinear filtering algorithm can be used to estimate the network parameters.

Referring to [30,31], the Elman network training algorithm based on SR-UKF for the state space model of Equation (9) can be concluded as Algorithm 1. Different from the reference [24], Equation (22) in Algorithm 1 adopts the Joseph stable form [32], which is numerically more stable because it can guarantee the symmetry and positive definite of

$P_{x_k}$ as long as $P_{x_k}^-$ is a symmetric positive definite matrix. The algorithm proposed in this study is concluded as Figure 2, and more details can be seen in Sections 3.2–3.4.

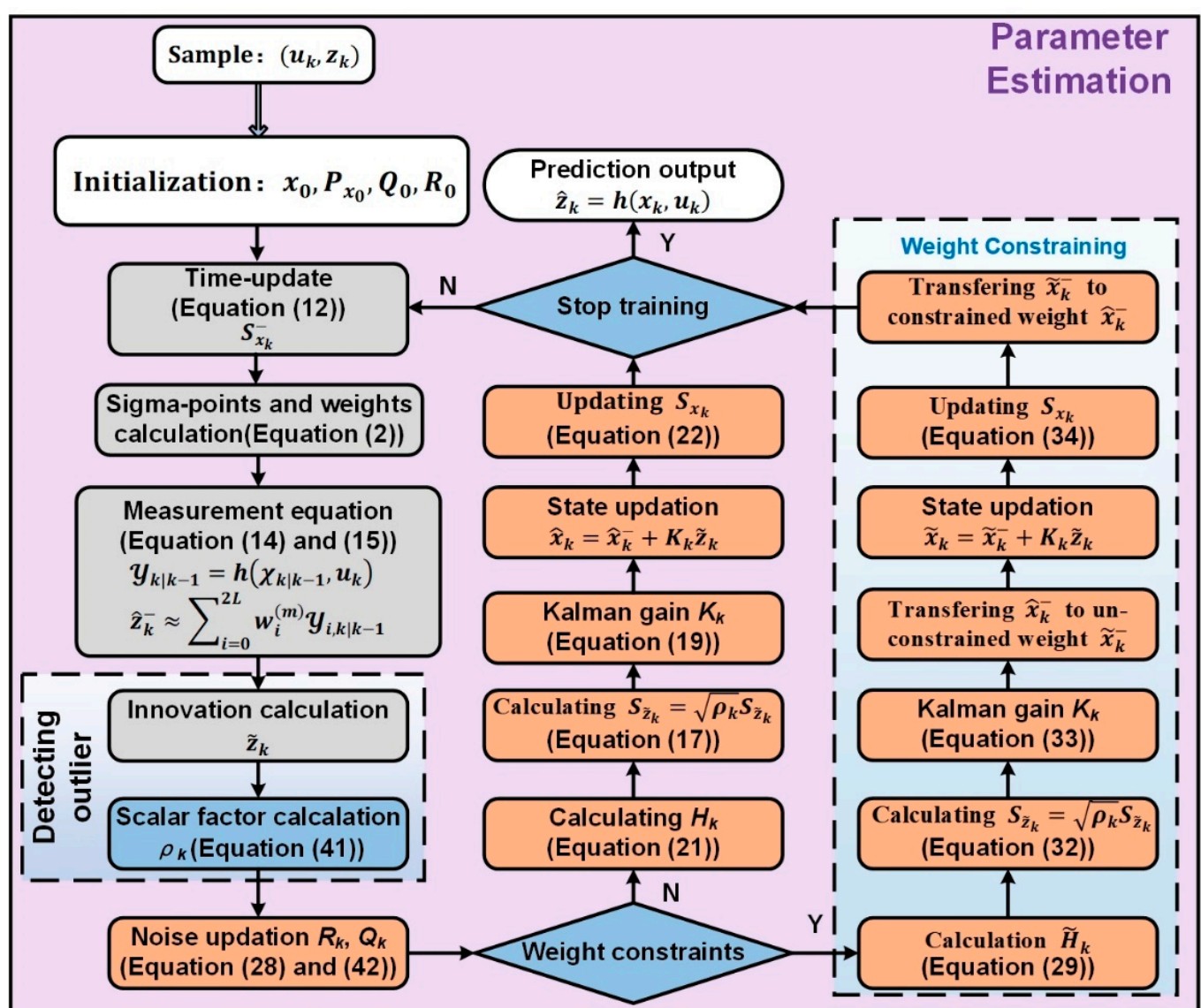

**Figure 2.** Schematic diagram of the proposed Elman-SR-UKF method for parameter estimation.

### 3.2. Update the Model with Adaptive Noise Variance

In fact, the accurate knowledge of the noise covariance required by the Kalman filter framework is often unknown and time-varying. Mismatching between the assumptive noise distribution and the actual noise distribution may degrade the prediction performance. To solve this problem, adaptive noise methods are often adopted to improve the accuracy and robustness of the state estimation. In this study, adaptive Sage-Husa noise estimation is proposed. In the proposed model, both measurement noise and process noise need to be properly estimated.

---

**Algorithm 1** Elman-SR-UKF algorithm

---

1:    Initialization:

2:    $\hat{x}_0 = E[x_0], \ \ P_{x_0} = E\left[(x - \hat{x}_0)(x - \hat{x}_0)^T\right], \ \ S_{x_0} = chol(P_{x_0})$      (10)

3:    For $k = 1, 2, \cdots$ ,

4:      Time update equations:

5:         $\hat{x}_k^- = \hat{x}_{k-1}$      (11)

6:         $\hat{x}_0 = E[x_0], \ \ P_{x_0} = E[(x - \hat{x}_0)(x - \hat{x}_0)^T], \ \ S_{x_0} = chol(P_{x_0})$      (12)

7:      Calculate sigma points:

8:         $\chi_{k|k-1} = \begin{bmatrix} \hat{x}_k^- & \hat{x}_k^- + \gamma S_{x_k}^- & \hat{x}_k^- - \gamma S_{x_k}^- \end{bmatrix}$      (13)

9:      Measurement update equations:

10:       $\mathcal{Y}_{k|k-1} = h\left(\chi_{k|k-1}, u_k\right)$      (14)

11:       $\hat{z}_k^- \approx \sum_{i=0}^{2L} w_i^{(m)} y_{i,k|k-1}$      (15)

12:       $S_{\tilde{z}_k} = qr\left\{ \left[ \ \sqrt{w_1^{(c)}}\left(\mathcal{Y}_{1:2L,k|k-1} - \hat{z}_k^-\right) \ \ S_{Q_k} \ \right] \right\}$      (16)

13:       $S_{\tilde{z}_k} = cholupdate\left\{ S_{\tilde{z}_k}, \mathcal{Y}_{0,k|k-1} - \hat{z}_k^-, w_0^{(c)} \right\}$      (17)

14:       $P_{x_k z_k} = \sum_{i=0}^{2L} w_i^{(c)}\left(\chi_{i,k|k-1} - \hat{x}_k^-\right)\left(\mathcal{Y}_{i,k|k-1} - \hat{z}_k^-\right)^T$      (18)

15:       $K_k = \left(P_{x_k z_k} / S_{\tilde{z}_k}^T\right) / S_{\tilde{z}_k}$      (19)

16:       $\hat{x}_k = \hat{x}_k^- + K_k\left(z_k - \hat{z}_k^-\right) = \hat{x}_k^- + K_k \tilde{z}_k$      (20)

17:       $H_k = P_{x_k z_k}^T \left(P_{x_k}^-\right)^{-1}, \ \ \ P_{x_k}^- = S_{x_k}^-\left(S_{x_k}^-\right)^T$      (21)

18:       $S_{x_k} = qr\left(\begin{bmatrix} S_{x_k}^- - K_k H_k S_{x_k}^- & K_k S_{Q_k} \end{bmatrix}\right)$      (22)

where $S_{x_k}$ and $S_{Q_k}$ are the square-root form of $P_{x_k}$ and $Q_k$, respectively, namely $P_{x_k} = S_{x_k} S_{x_k}^T$, $Q_k = S_{Q_k} S_{Q_k}^T$. $R_k$ is the process noise covariance. $\gamma = \sqrt{L + \lambda}$ is a composite scaling parameter. $z_k$ and $\tilde{z}_k$ are the true value and innovation at time step $k$ respectively. The $diag\{\cdot\}$ operator zeros all the elements of a square matrix except the main diagonal. $qr\{\cdot\}$ and $cholupdate\{\cdot\}$ are standard MATLAB functions, representing QR decomposition and Cholesky factor updating, respectively. The operator "/" stands for the right division operation of MATLAB. The origin of Equation (22) can be referred to the derivation of Equation (42).

---

### 3.2.1. Adaptive Noise Estimation

Currently, the adaptive filtering algorithm is a common adaptive noise method [33]. Assuming that the residual variable $\tilde{z}_k$ is stable, the residual covariance can be approximated by the residual variable within the moving window:

$$P_{\tilde{z}_k \tilde{z}_k} = S_{\tilde{z}_k} S_{\tilde{z}_k}^T \approx \frac{1}{N} \sum_{j=0}^{N-1} \tilde{z}_{k-j} \tilde{z}_{k-j}^T \tag{23}$$

where $\tilde{z}_k = z_k - \hat{z}_k^-$ is the residual variable, $z_k$ is the measured value and $\hat{z}_k^-$ is the predicted value which is given in Equation (15). $N$ is the width of the moving window. More details on how to optimize window width can be found in [34].

Combining Equation (16) with Equation (23), the estimation of the measurement noise covariance $Q_k$ can be obtained as follows:

$$\overline{Q}_k^* = P_{\tilde{z}_k \tilde{z}_k} - P_{k|k-1}^{zz}, \quad P_{k|k-1}^{zz} = \sum_{i=0}^{2L} w_i^{(m)}\left(\mathcal{Y}_i - \hat{z}_k^-\right)\left(\mathcal{Y}_i - \hat{z}_k^-\right)^T \tag{24}$$

Considering that $\overline{Q}_k^*$ may not be a positive definite matrix, the following rule [35] is recommended:

$$\overline{Q}_k = diag\left\{ \left|\overline{Q}_k^*(1)\right|, \left|\overline{Q}_k^*(2)\right|, \cdots, \left|\overline{Q}_k^*(m)\right| \right\} \tag{25}$$

where $\overline{Q}_k^*(i)$ is the $i$-th diagonal element of the matrix $\overline{Q}_k^*$, and the abovementioned formula can ensure the positive definiteness of $\overline{Q}_k$.

Similarly, the estimation of the process noise covariance matrix $R_k$ can be written as:

$$\overline{R}_k = K_k \left( \frac{1}{N} \sum_{j=0}^{N-1} \tilde{z}_{k-j}\tilde{z}_{k-j}^T \right) K_k^T \tag{26}$$

### 3.2.2. Adaptive Sage-Husa Noise Estimation

Taking into account the uncertainty of process noise, to compensate for the error caused by the variations of noise statistics and estimate the process noise covariance, this study further adopts the Sage-Husa estimator [36] to estimate and adjust the measurement noise covariance.

$$\hat{Q}_k = (1 - d_{k-1})\hat{Q}_{k-1} + d_{k-1}\overline{Q}_k \tag{27}$$

where $\overline{Q}_k$ is defined as Equation (25), $d_{k-1} = \frac{1-b}{1-b^k}$, $b$ is the forgetting factor and $0.95 \leq b \leq 0.995$.

Similarly, the adaptive process noise covariance can be obtained:

$$\hat{R}_k = (1 - d_{k-1})\hat{R}_{k-1} + d_{k-1}\overline{R}_k \tag{28}$$

The definition of $\overline{R}_k$ is shown in Equation (26).

The adaptive Sage-Husa noise combines the noise covariance at the previous moment and the estimated noise covariance, which is then recursively able to estimate the unknown time-varying covariance and to achieve better robustness and accuracy.

### 3.3. Handling Outliers

Sensor measurements may suffer from outliers due to abnormal conditions, which easily cause the deviation of adaptive covariance from the true distribution and result in a decrease in prediction performance. Therefore, addressing outliers is imperative for accurate prediction. The abovementioned calculation shows that both the estimation of noise and the update of the state vector are directly related to the residual error. Using residual error for outlier identification is an intuitive idea.

### 3.3.1. Outliers Identification

This study adopted hypothesis testing to detect abnormal conditions using statistical information of residual error [33]. The statistic $\alpha_k$ are defined as:

$$\alpha_k = \tilde{z}_k^T \left( H_k P_{x_k}^- H_k^T + R_k \right)^{-1} \tilde{z}_k \tag{29}$$

The statistic $\alpha_k$ is assumed to follow the distribution of $\chi^2$ with degree-of-freedom $s$, where $s$ is the dimension of the residual error vector $\tilde{z}_k$. Choosing the significance level $\alpha_\chi (0 \leq \alpha_\chi \leq 1)$, $\chi^2_{\alpha,s}$ can be determined by

$$P\left( \chi^2 > \chi^2_{\alpha,s} \right) = \alpha_\chi \tag{30}$$

If the alternative hypothesis $H_1$ holds, the statistic $\alpha_k$ is greater than the threshold, which can be represented as:

$$\begin{aligned} H_0 &: \alpha_k \leq \chi^2_{\alpha,s} \quad \forall k; \\ H_1 &: \alpha_k > \chi^2_{\alpha,s} \quad \exists k. \end{aligned} \tag{31}$$

### 3.3.2. Parameter Adjustment

This article introduced a scalar factor $\rho_k$ to adjust the measurement error covariance and ensure the robustness of system [37]:

$$P_{\tilde{z}_k\tilde{z}_k} = \rho_k P_{\tilde{z}_k\tilde{z}_k} \tag{32}$$

where $\rho_k$ can be calculated as:

$$\rho_k = \begin{cases} 1 & , \alpha_k \leq \chi^2_{\alpha,s} \\ \frac{\alpha_k}{\chi^2_{\alpha,s}} & , \alpha_k > \chi^2_{\alpha,s} \end{cases} \tag{33}$$

At the same time, the calculation of $\overline{Q}^*_k$ in Equation (25) should be modified as:

$$\overline{Q}^*_k = (\rho_k - 1)P^{zz}_{k|k-1} - \rho_k \hat{Q}_{k-1} \approx \frac{\rho_k - 1}{N} \sum_{j=0}^{N-1} \tilde{z}_{k-j}\tilde{z}^T_{k-j} + \rho_k \hat{Q}_{k-1} \tag{34}$$

When the abnormal values are detected by Equation (31), i.e., the alternative hypothesis $H_1$ holds, the scale factor $\rho_k$ amplifies the observation error covariance, which leads to a decrease in the Kalman filter gain, and then to reduce the magnitude of the state update and improve the robustness of the filter. A specific summary of the Elman-based the SR-UKF algorithm proposed in this study is shown in Figure 2.

### 3.4. Weight Constraining

To limit the range of weight values and to ensure that the Elman achieves smooth mapping, this study introduces a heuristic method [38] to constrain the weights.

In essence, this process has a similar function as the Bayesian regularization method, which prevents model overfitting and keeps the model smoother. However, this constrained algorithm may lead to a degradation of performance.

The main principle of the weight constraining algorithm is as follows: Consider the mapping $x^{i,j}_k = \phi\left(\tilde{x}^{i,j}_k, \mu\right)$, which transforms $\tilde{x}^{i,j}_k \in [-\infty, +\infty]$ to the constraint weight space $x^{i,j}_k \in [-\mu, +\mu]$, and the mapping $\tilde{x}^{i,j}_k = \phi^{-1}\left(x^{i,j}_k, \mu\right) \in [-\infty, +\infty]$ has the opposite effect. The function $\phi\left(\tilde{x}^{i,j}_k, \mu\right)$ transfers the weight from an unconstrained space to a constrained space and must meet the following properties:

(1)  $\phi$ is a continuously differentiable function on $[-\infty, +\infty]$;
(2)  $\phi^{-1}$ exists and is a continuous function $[-\mu, +\mu]$;
(3)  $\lim_{\mu \to \infty} \phi\left(\tilde{x}^{i,j}_k, \mu\right) = \tilde{x}^{i,j}_k = x^{i,j}_k$.

The weight update is performed with Equations (19)–(22) in the unconstrained weight space. Once the update is completed, the unconstrained weights are converted back to the constrained space.

To perform SR-UKF recursion in the unconstrained space, some modifications are needed. For Equation (21), $H_k$ can be interpreted as the derivative of output $z_k$ with respect to the weight vector $x_k$ of each node in Elman-SR-UKF. The derivative of the weights in the constrained space must be converted to the derivative of the weights in the unconstrained space, and then used to update the weights in the unconstrained space. This can be achieved by the following transformation:

$$\widetilde{H}_k = \begin{bmatrix} \frac{\partial z^1_k}{\partial x^1_k}\frac{\partial x^1_k}{\partial \tilde{x}^1_k} & \cdots & \frac{\partial z^1_k}{\partial x^L_k}\frac{\partial x^L_k}{\partial \tilde{x}^L_k} \\ \vdots & \ddots & \vdots \\ \frac{\partial z^s_k}{\partial x^1_k}\frac{\partial x^1_k}{\partial \tilde{x}^1_k} & \cdots & \frac{\partial z^s_k}{\partial x^L_k}\frac{\partial x^L_k}{\partial \tilde{x}^1_k} \end{bmatrix}_{s \times L} = \begin{bmatrix} \frac{\partial z^1_k}{\partial x^1_k} & \cdots & \frac{\partial z^1_k}{\partial x^L_k} \\ \vdots & \ddots & \vdots \\ \frac{\partial z^s_k}{\partial x^1_k} & \cdots & \frac{\partial z^s_k}{\partial x^L_k} \end{bmatrix} \begin{bmatrix} \frac{\partial x^1_k}{\partial \tilde{x}^1_k} & & \\ & \ddots & \\ & & \frac{\partial x^L_k}{\partial \tilde{x}^1_k} \end{bmatrix} = H_k \Lambda_W \approx P^T_{x_k z_k}\left(P^-_{x_k}\right)^{-1} \Lambda_W \tag{35}$$

where $s$ and $L$ represent the dimensions of the output and the state vector, respectively. $z^i_k$ and $x^i_k$ are the $i$-th component of the output $z_k$ and the constraint weight vector $x_k$, respectively. The conversion formula between the unconstrained weight $\tilde{x}^{i,j}_k$ and the constraint weight $x^{i,j}_k$ can be chosen as Equations (36) and (37).

$$x_k^{i,j} = \phi\left(\widetilde{x}_k^{i,j}, \mu\right) = \frac{\widetilde{x}_k^{i,j}}{1 + \left|\widetilde{x}_k^{i,j}\right|/\mu} \tag{36}$$

$$\widetilde{x}_k^{i,j} = \phi^{-1}\left(x_k^{i,j}, \mu\right) = \frac{x_k^{i,j}}{1 - \left|x_k^{i,j}\right|/\mu} \tag{37}$$

According to the abovementioned formulas, when $\mu \to \infty$, $\widetilde{x}_k^{i,j} \to x_k^{i,j}$. Additionally, when $x_k^{i,j} \to \mu$, $\widetilde{x}_k^{i,j} \to \infty$.

The Elman-SR-UKF algorithm with constrained weights needs to rewrite $S_{\widetilde{z}_k}$, $K_k$ and $S_{x_k}$ in Algorithm 1 as follows:

$$S_{\widetilde{z}_k} = qr\left\{\begin{bmatrix} \widetilde{H}_k S_{x_k}^- & S_{Q_k} \end{bmatrix}\right\} \tag{38}$$

$$K_k = P_{x_k}^- \widetilde{H}_k^T / \widetilde{S}_{\widetilde{z}_k}^T / \widetilde{S}_{\widetilde{z}_k}, \quad P_{x_k}^- = S_{x_k}^- S_{x_k}^{-T} \tag{39}$$

$$S_{x_k} = qr\left\{\begin{bmatrix} S_{x_k}^- - K_k \widetilde{H}_k S_{x_k}^- & K_k S_{Q_k} \end{bmatrix}\right\} \tag{40}$$

From Equation (39), Equation (40) can be calculated as follows:

$$K_k P_{\widetilde{z}_k \widetilde{z}_k} = P_{x_k}^- \widetilde{H}_k^T \tag{41}$$

Combining Equations (19) and (41), $P_{x_k}$ can be rewritten as:

$$
\begin{aligned}
P_{x_k} &= P_{x_k}^- - P_{x_k}^- \widetilde{H}_k^T K_k^T + K_k P_{\widetilde{z}_k \widetilde{z}_k} K_k^T - K_k \widetilde{H}_k (P_{x_k}^-)^T \\
&= S_{x_k}^-\left(S_{x_k}^-\right)^T - S_{x_k}^-\left(S_{x_k}^-\right)^T \widetilde{H}_k^T K_k^T - K_k \widetilde{H}_k \left(S_{x_k}^-\left(S_{x_k}^-\right)^T\right)^T + K_k\left(\widetilde{H}_k S_{x_k}^-\left(S_{x_k}^-\right)^T \widetilde{H}_k^T + S_{Q_k} S_{Q_k}^T\right) K_k^T \\
&= \begin{bmatrix} S_{x_k}^- - K_k \widetilde{H}_k S_{x_k}^- & K_k S_{Q_k} \end{bmatrix} \times \begin{bmatrix} S_{x_k}^- - K_k \widetilde{H}_k S_{x_k}^- & K_k S_{Q_k} \end{bmatrix}^T
\end{aligned} \tag{42}
$$

From the abovementioned formula, Equation (40) can be finally obtained.

## 4. Evaluation Methods and Validation Materials

The main purpose of this section was to provide the basic knowledge for method evaluations. Section 4.1 offers the evaluation indicators to assess how good the prediction performance of a model is. The following section shows the materials for the case study.

### 4.1. Evaluation Methods

A dataset collected from a real wastewater treatment plant is applied to evaluate the performance of the proposed soft-sensor model. At the same time, the Elman-SR-UKF model is compared with the traditional Elman network models to verify the prediction performance of the important indicators of sewage. To evaluate the prediction accuracy of the model, the root mean square error (RMSE), mean relative error (MRE), and correlation coefficient (R) of Equations (43) and (45) are introduced accordingly. Their mathematical expressions are given below:

$$RMSE = \sqrt{\frac{\sum_{i=0}^n (\hat{y}_i - y_i)^2}{n}} \quad, i = 1, 2, \cdots n \tag{43}$$

$$MAE = \frac{\sum_{i=0}^n (|\hat{y}_i - y_i|)}{\hat{y}_i} \tag{44}$$

$$R(Y, \hat{Y}) = \frac{\text{cov}(Y, \hat{Y})}{\sqrt{\text{var}(Y)\text{var}(\hat{Y})}} \tag{45}$$

where $n$ is the number of samples, $\hat{y}_i$ and $y_i$ are the predicted value and the real value respectively, and $\hat{Y} = (\hat{y}_1, \hat{y}_2, \cdots, \hat{y}_n)$, $Y = (y_1, y_2, \cdots, y_n)$. $\mathrm{cov}(Y, \hat{Y})$ is the covariance of $Y$ and $\hat{Y}$, and $\mathrm{var}(Y)$ and $\mathrm{var}(\hat{Y})$ are the variance of $Y$ and $\hat{Y}$, respectively.

Since the multioutput model predicts multiple responses at one time, the root mean of diagonal square sum (RMSSD) and multiple correlation coefficient (MR) are used as the performance evaluation criteria of multioutput. The formulas for these evaluation criteria are as follows:

$$RMSSD = \sqrt{\frac{1}{N} trace\left\{ (Y - \hat{Y})^T (Y - \hat{Y}) \right\}} \tag{46}$$

$$MR = \frac{1}{l} \sum_{i=1}^{l} \left| R^i \right| \tag{47}$$

where trace represents the *trace* of the matrix, and $R^i$ represents the correlation coefficient of the $i$-th output. A smaller RMSSD and a larger MR mean better prediction performance for a soft sensor.

From the standpoint of monitoring performance, the indicators of RMSE, R and MAE are used to evaluate the prediction accuracy and deviation ratio of each output variable while the RMSSD and MR are employed to evaluate the overall predictive performance of the multioutput model.

### 4.2. Materials for the Case Study

The case study for validation is a real wastewater treatment plant. The collected dataset was sampled at daily intervals in an urban wastewater treatment plant. The total number of data covers approximately 2 years [39].

As shown in Figure 3, the wastewater plant process consists of four components: pretreatment, primary settler, aeration tanker and secondary settler. Partial sludge in the secondary settler is recycled back to the aeration tank to maintain the number of microorganisms in the aeration tank, and the other part of the useless sludge is discharged as wasted sludge. Due to the lack of the instrument, the dataset with 38 process variables is collected once a day. Additional details are provided in [40,41].

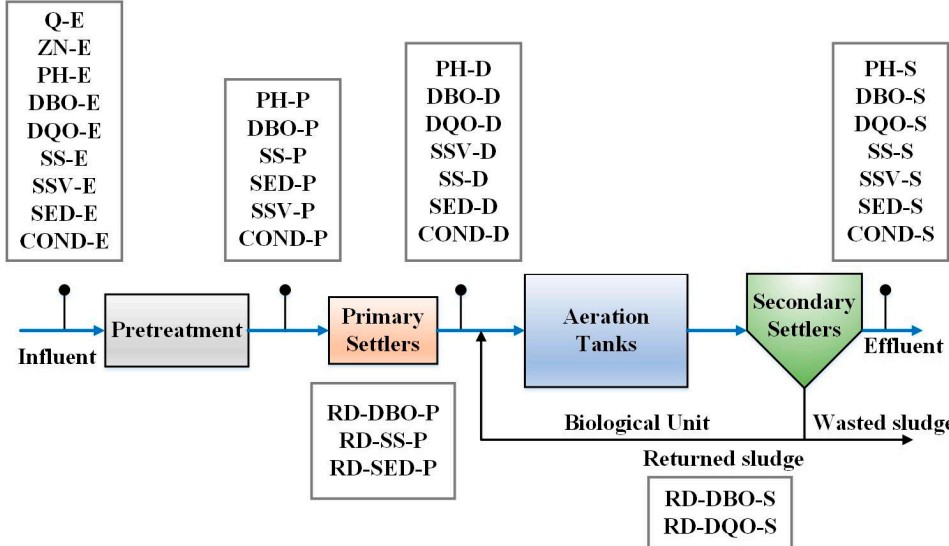

**Figure 3.** Process of wastewater treatment.

After preliminary screening, 18 process variables are selected as inputs of the model, which are shown in Table 1. This study selects the output chemical demand of oxygen (DQO-S), output biological demand of oxygen (DBO-S), and output suspended solids (SS-S) in the effluent of the plant as output variables. During model training, the first 200 samples are utilized for training, and the remaining 200 data points are used as a test set.

**Table 1.** Selected input variables for soft-sensor modeling.

| No | Variables | Comments |
|---|---|---|
| 1 | DBO-E | Input biological demand of oxygen to plant |
| 2 | DQO-E | Input chemical demand of oxygen to plant |
| 3 | DBO-P | Input biological demand of oxygen to primary settler |
| 4 | PH-D | Input pH to secondary settler |
| 5 | DBO-D | Input biological demand of oxygen to secondary settler |
| 6 | DQO-D | Input chemical demand of oxygen to secondary settler |
| 7 | SS-D | Input suspended solids to secondary settler |
| 8 | SED-D | Input sediments to secondary settler |
| 9 | RD-DBO-P | Performance input biological demand of oxygen in primary settler |
| 10 | RD-SS-P | Performance input suspended solids to primary settler |
| 11 | RD-DBO-S | Performance input biological demand of oxygen to secondary settler |
| 12 | RD-DQO-S | Performance input chemical demand of oxygen to secondary settler |
| 13 | RD-DBO-G | Global performance input biological demand of oxygen |
| 14 | RD-DQO-G | Global performance input chemical demand of oxygen |
| 15 | RD-SS-G | Global performance input suspended solids |
| 16 | RD-SED-G | Global performance input sediments |
| 17 | PH-S | pH in the effluent |
| 18 | SED-S | Sediments in the effluent |

## 5. Results and Discussion

First, the predicted performance was assessed by comparing the Elman-SR-UKF (Elman network based on square-root unscented Kalman filter algorithm) with the traditional Elman networks (Elman-BPTT, Elman-GDM, Elman-LM, and Elman-RTRL-LM) based on different classical training algorithms, i.e., BPTT (back propagation through time algorithm), GDM (Momentum Gradient Descent algorithm), LM (Levenberg-Marquardt algorithm) and RTRL-LM (https://github.com/yabata/pyrenn, accessed on 20 November 2021) (real-time recurrent learning based on Levenberg-Marquardt algorithm). Then, an Elman network based on SR-UKF with different constrained weights was implemented and compared for quality-related variable prediction in the wastewater plant. In this paper, the extreme learning machine (ELM) [42] and multi-output Gaussian process regression (MGPR) [43] are further introduced to act as the baselines for comparisons.

A wastewater treatment system is a physical, chemical, and biological system that makes it multivariable, nonlinear, and dynamic. Generally, it is necessary to monitor multiple variables of the wastewater treatment process simultaneously. Due to the strong coupling and high interaction among the output variables, using a set of independent single-output models to predict multiple quality-related output variables is inadequate. From the perspective of practical application, this study proposed a multioutput neural network model, Elman-SR-UKF, which can deal with multivariate problems. In this case, the DQO-S, DBO-S, and SS-S in the UCI data are selected as the prediction variables.

To verify the proposed model in this study, an 18-8-3 Elman network architecture is constructed, while the activation functions $f_h(\cdot)$ of the hidden layer and output activation functions $f_o(\cdot)$ are designed as ***logsig*** functions and ***purelin*** functions, respectively. The initial value of weights is a random number between $[-0.5, 0.5]$. Equations (43)–(45) are used as evaluation criteria for predictive performance. The parameter definitions of the model are shown in Table 2. The software used in this study was MATLAB R2016a.

**Table 2.** Parameters definition of the model.

| Models | Parameters |
|---|---|
| Elman-SR-UKF | $\alpha = 1$, $\beta = 0$, $\kappa = 2$, forgetting factor $b = 0.955$, initial process covariance $R_0 = 1 \times 10^{-5}I$, initial measurement covariance $Q_0 = 0.5I$, initial error covariance $P_{x_0} = 0.01I$, moving window $N = 20$, statistic $\alpha_\chi = 0.05$ |
| Other Elman models | learning rate lr = 0.0l, iteration = 1000 |

In Section 3.1, this study mentioned that the parameter $\alpha$ should ideally be a small number. However, if $\alpha$ is too small, the weights of the sigma points are all negative, which may easily lead to a diversion problem in the iteration. In this study, $\alpha$ was set to 1 to ensure the positive weights of sigma points. ***I*** represents the identity matrix. Taking into account the randomness of the initial weights, the results in Tables 3 and 4 are the average results obtained by running 500 models with random initial weights. In Table 3, $\mu = \infty$ means that there is no weight constraining. Because the effect of the RTRL algorithm is nonideal, in comparison, this study uses the RTRL-based LM algorithm in the Pyrenn toolbox. Epochs represent the number of iterations required for convergence.

**Table 3.** RMSE, R, RMSSD, MR, and epochs values of the output variables.

| Models | Predicted Variables | SS-S | DBO-S | DQO-S | RMSSD | MR | Epochs |
|---|---|---|---|---|---|---|---|
| Elman-BPTT | RMSE | 4.957 | 3.591 | 16.125 | 17.271 | 0.688 | 1000 |
| | R | 0.622 | 0.709 | 0.732 | | | |
| | MRE | 0.246 | 0.172 | 0.190 | | | |
| Elman-GDM | RMSE | 4.598 | 3.308 | 14.479 | 15.569 | 0.781 | 1000 |
| | R | 0.705 | 0.799 | 0.834 | | | |
| | MRE | 0.224 | 0.154 | 0.166 | | | |
| Elman-LM | RMSE | 25.961 | 3.3581 | 13.531 | 31.356 | 0.7648 | 975 |
| | R | 0.471 | 0.906 | 0.917 | | | |
| | MRE | 1.273 | 0.145 | 0.135 | | | |
| Elman-RTRL-LM | RMSE | 21.184 | 2.434 | 10.142 | 24.529 | 0.770 | 80 |
| | R | 0.457 | 0.922 | 0.931 | | | |
| | MRE | 0.870 | 0.092 | 0.091 | | | |
| Elman-SR-UKF ($\mu = \infty$) | RMSE | **3.330** | **1.765** | **8.028** | **8.847** | **0.917** | **30** |
| | R | **0.857** | **0.946** | **0.949** | | | |
| | MRE | **0.151** | **0.076** | **0.081** | | | |
| ELM | RMSE | 5.773 | 4.007 | 18.065 | 19.431 | 0.626 | / |
| | R | 0.523 | 0.664 | 0.690 | | | |
| | MRE | 0.278 | 0.193 | 0.205 | | | |
| MGPR | RMSE | 4.412 | 3.057 | 13.012 | 14.121 | 0.807 | / |
| | R | 0.742 | 0.826 | 0.854 | | | |
| | MRE | 0.213 | 0.141 | 0.139 | | | |

Bold: the best results.

Table 3 shows the results of different algorithms and the values in bold represent the best results. By comparing all criterion evaluation results of the models, it can be seen that, on average, Elman-SR-UKF achieved the best prediction performance with the smallest RMSE, MRE, and R for all three outputs. The MRE represents the ratio of the absolute difference of the measurement to the actual measurement. The smallest MRE of Elman-SR-UKF for approximately 15% in SS-S and 8% in DBO-S and DQO-S can obviously represent that Elman-SR-UKF the best monitoring method among the abovementioned methods. It is clear that the RMSSD of Elman-SR-UKF is 95.3%, 76%, 254%, 177%, 119.6%, and 59.6% was better than those of Elman-BPTT, Elman-GDM, Elman-LM, Elman-RTRL-LM, ELM, and MGPR, respectively. MR of Elman-SR-UKF exhibited similar profiles.

MGPR achieved a relatively good result in terms of RMSSD and MR. However, by comparing the RMSE, R, and MRE, MGPR had a worse performance in the outputs of DBO-S and DQO-S than Elman-RTRL-LM and Elman-LM. That is because MGPR was not able to learn non-stationary process properly and some variables in the dataset were relatively stable but still did not completely follow a stationary process. The performance of ELM in this paper was not good. This is mainly because the pre-fixed input weights in ELM limited the representation capability of the model and usually could not be used for complex tasks.

The results of RMSSD and MR reflect that Elman-SR-UKF can perform well from the standpoint of the overall predictive performance of the multioutput model, which

illustrates that the proposed method can exhibit excellent performance and overcome the underprediction phenomena in traditional Elman network.

In terms of convergence, Elman-SR-UKF had the fastest convergence speed, and the convergence of the Elman-RTRL-LM algorithm was faster than that of the Elman-BPTT and Elman-GDM algorithms. The convergence rates of Elman-LM and Elman-SR-UKF were faster than those of the gradient descent algorithms, such as Elman-BPTT and Elman-GDM. The Elman-LM algorithm utilized the second-order quasi-Newton optimization method to train the network, and the Elman-SR-UKF algorithm used the covariance of the second-order statistical property of variables for iteration, which indicates that the second-order method may have a faster convergence than the first-order method. Moreover, the Elman-SR-UKF algorithm had better convergence performance, which proves the effectiveness of the proposed method.

The prediction profiles of different models for output variables are further shown in Figure 4. To clarify the prediction profiles, only the first 80 time series data in the testing dataset were shown. By comparing the prediction results of various methods, it can be seen that the Elman-SR-UKF algorithm can better track the dynamic variations of output variable SS-S, while the fitting between the predicted and real values with respect to other algorithms about SS-S was poor. This occurs mainly because the algorithm fell into a local minimum, which, in turn, proved the effectiveness of the proposed method in this study. For the variables, DBO-S and DQO-S, the Elman-LM, Elman-RTRL-LM, and Elman-SR-UKF methods had good fitting performance. In terms of the prediction results of peaks and valleys in the prediction profiles, Elman-SR-UKF had excellent prediction performance in tracking the dynamic variations of the targets and its prediction ability was better than the traditional Elman training algorithm listed above, which shows the superiority of the Elman-SR-UKF in this study.

Elman-SR-UKF assumes the parameters of the Elman network to be state random variables, and then recursively updates the posterior density of the state to optimize an instantaneous cost function. In addition, the inherent statistical averaging of the Elman-SR-UKF algorithm can be less likely to get stuck in local minima, which makes the Elman-SR-UKF able to achieve the best performance. Additionally, it is important to note that Elman-GDM achieved better performance, which is mainly because Elman-GDM added the past gradient information to the parameter update equation and still optimized the cost function toward the gradient direction. Therefore, the gradient may change a lot over relatively small regions in the search space and jump out of the local minima.

Table 4 shows the prediction results of output variables with different weight constraints in terms of RMSE, MRE, R, RMSSD, and MR. From the perspective of the maximum weight, constraints can be successfully restrained in the range of weight values, which can prevent the overfitting of the model. Nevertheless, from the results of RMSE, MRE, R, RMSSD, and MR, it can be found that weight constraints have some effects on prediction performance. This problem can result from two aspects: First, setting up constraints based on weight values makes the model less sensitive to the error. Conversely, if there is no weight constraint, it means that the model can adapt to measurement error and exhibit oversensitive behaviors, which may, in turn, bring better performance and faster convergence. However, a sensitive model with unconstrained weights may produce unstable results, thus reducing the reliability of the soft sensor model. It is envisioned that the model without weight constraints can achieve better results for a stationary dataset. In this actual activated sludge water plant, once the outliers have been preprocessed, the stationary requirement for a dataset can be guaranteed. Second, once the constrained weight values reach the saturation region, the derivative of the constrained weight with respect to the unconstrained weight becomes zero, making it difficult to perform further training with these particular weight values. In summary, weight constraints may lead to a slight degradation in performance, but its performance is still better than that of the Elman-BPTT, Elman-GMD, Elman-LM and Elman-RTRL-LM methods. Moreover, the weight constraining algorithm

can better meet the parameter requirements in applications, such as fixed-point arithmetic, and improve the reliability of a soft sensor.

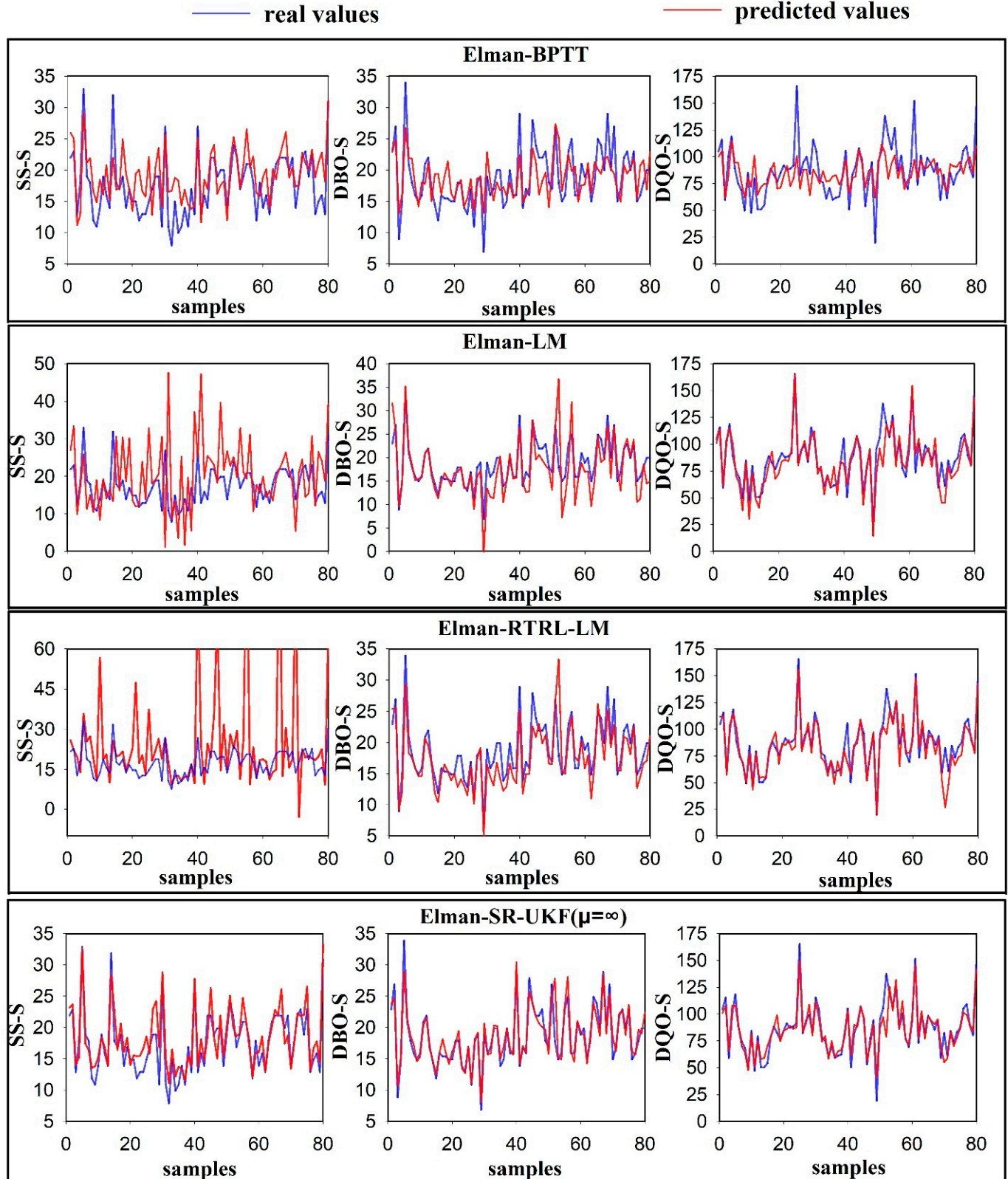

**Figure 4.** Prediction profiles of output variables compared with real values. (The first 80 time series data in the testing dataset).

**Table 4.** RMSE, R, RMSSD, and MR values of the output variables with different weight constraints ($P_{x_0} = 0.05I$).

| Elman-SR-UKF with Constrained Weights | Predicted Variables | SS-S | DBO-S | DQO-S | RMSSD | MR | Maximum Weight |
|---|---|---|---|---|---|---|---|
| $\mu = 1$ | RMSE | 3.967 | 2.053 | 8.701 | 17.271 | 0.688 | 0.80 |
| | R | 0.792 | 0.927 | 0.939 | | | |
| | MRE | 0.181 | 0.091 | 0.089 | | | |
| $\mu = 2$ | RMSE | 3.646 | 1.894 | 8.805 | 9.625 | 0.903 | 1.303 |
| | R | 0.834 | 0.938 | 0.938 | | | |
| | MRE | 0.168 | 0.082 | 0.089 | | | |
| $\mu = 5$ | RMSE | 3.596 | 1.851 | 8.666 | 9.507 | 0.908 | 2.15 |
| | R | 0.841 | 0.942 | 0.941 | | | |
| | MRE | 0.165 | 0.079 | 0.086 | | | |
| $\mu = \infty$ | RMSE | **3.563** | **1.802** | **8.26** | **9.177** | **0.912** | 4.12 |
| | R | **0.847** | **0.945** | **0.946** | | | |
| | MRE | 0.160 | 0.076 | 0.080 | | | |

Figure 5 shows the convergence profiles of different weight constraints with $P_{x_0} = 0.05I$. As shown in Figure 5, the convergence speed with unconstrained weights is faster than that with constrained weights, which can be attributed to the lower sensitivity to error. Although weight constraints slightly affect the convergence speed, the algorithm still converges at a relatively fast iterative speed.

The above results show that the Elman-SR-UKF can create more accurate and robust results, thus overcoming the underprediction phenomena in data-driven process monitoring [44]. Although the main principle of this paper is different from the reference [44], it is very easy to extend this study to the quantify the uncertainty of effluent variables forecasting. Compared with the application of Li et al. [45,46], this paper provides a better prediction performance and a simple sequential way to update the parameters online. Thus, this study can be successfully envisioned to be applied to process monitoring of effluent variables in wastewater plants.

In summary, according to all prediction results of the models, the Elman-SR-UKF method can perform more accurate prediction results and has a faster convergence. Of note, for the output variable SS-S, the Elman-SR-UKF can better fit the target variations, which further demonstrates the effectiveness of the proposed algorithm. From the perspective of weight constraints, the weight constraining method can effectively restrict the range of the weight values and reduce the sensitivity to the error, which can, in turn, improve the reliability of a soft sensor but may slightly affect the convergence speed and prediction performance of the model. Moreover, the prediction performance of the method with a constrained algorithm is still better than that of the Elman-BPTT, Elman-GMD, Elman-LM and Elman-RTRL-LM methods.

**Remark 1.** *In this case study, an activated sludge-based treatment process is used for validation, which is the most commonly used treatment process around the world and covers more than 80% of WWTPs in China. However, the physical, chemical, and biological phenomena associated with treatment units (primary classifier, anaerobic, anoxic, aerobic tanks and secondary classifier) always add significant complexity to process monitoring and plant management. Moreover, the inter- and intercorrelation for the reactors and processes make the collected data exhibit strong nonlinearity and high dynamics among the variables. This renders it necessary to use adaptive and nonlinear data driven models for prediction, such as recursive neural networks. However, general RNNs cannot take into account uncertainties inside and outside WWTPs. (a) Biomass growth, death, and nutrient consumption are all sensitive to many factors, such as pH and temperature. (b) The large variety of biological species present in WWTPs makes it impossible to accurately determine all kinetic parameters with a general model. (c) The organic loads in the influent significantly fluctuate depending on the level of human activity and external environment. For instance, in urban WWTPs, the organic loads vary during the day according to the level of human activity. Additionally, they*

*themselves are strongly affected by weather conditions and seasonal change. Kalman filters are inherently dynamic algorithms with the ability to describe uncertainties. To take into account all aforementioned uncertainties, Elman works together with Kalman filters to make predictions for quality-related but hard-to-measure effluent variables. An Elman network together with Kalman filters is also able to capture the aforementioned nonlinear relationship.*

**Remark 2.** *In this case study, the proposed soft sensor is only used for effluent quality prediction in an activated sludge process. However, there are still many hard-to-measure variables in the entire WWTPs, for example, sludge volume index (SVI) in the secondary classifier, toxic loads in the influent and $N_2O$ during the reaction. In addition to activated sludge processes, the proposed methods can be extended and used for other processes such as oxidation ditches (ODs) and sequencing bath reactors (SBRs).*

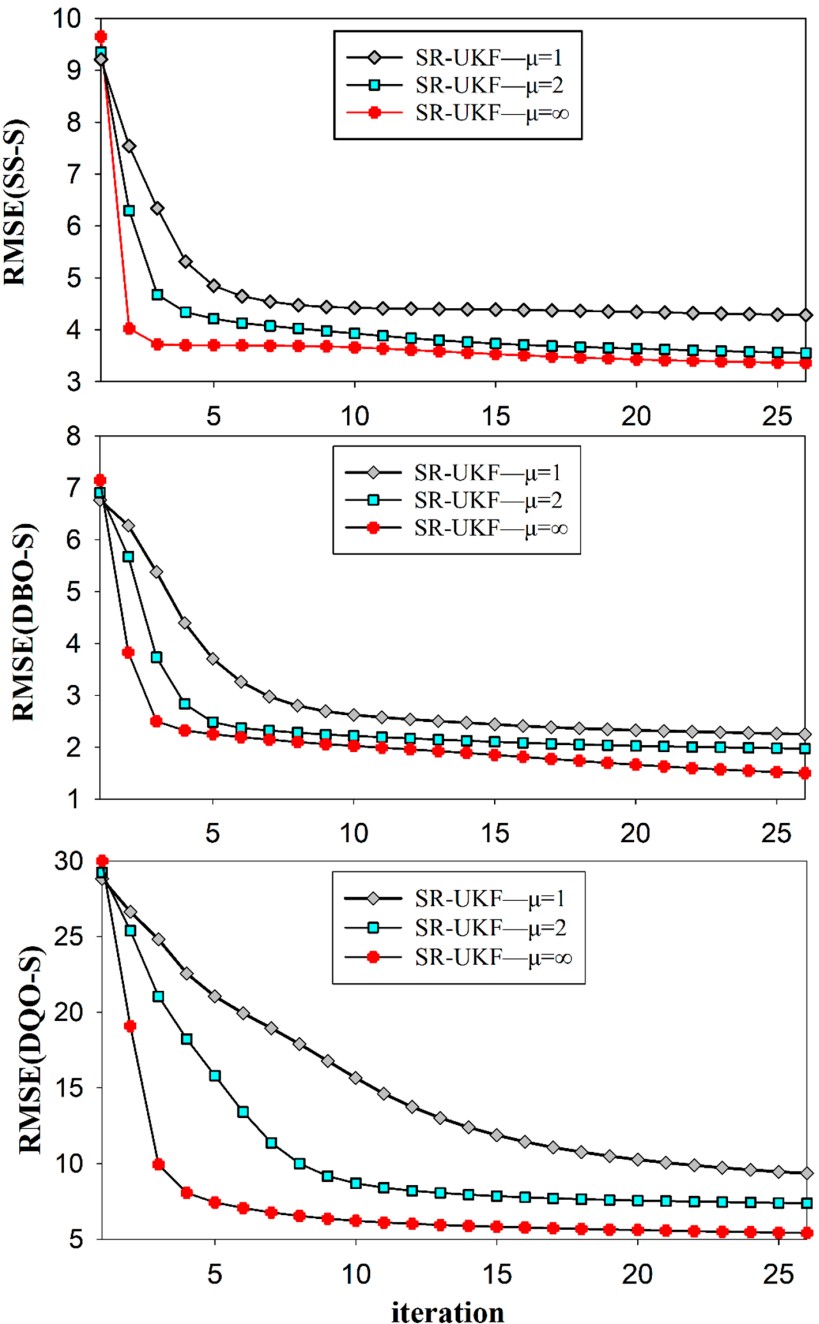

**Figure 5.** Convergence profiles of different weight constraints ($P_{x_0} = 0.05I$).

## 6. Conclusions

To address the highly nonlinear and dynamic prediction issues in wastewater treatment plants and to address the problems of slow convergence speed and local minima in standard Elman neural networks, this study proposed a novel soft-sensor model with an Elman neural network trained by the proposed SR-UKF algorithm, aiming to sense the quality-related variables in wastewater treatment. In the proposed method, residual-based Sage-Husa adaptive noise was adopted to solve the problem of unknown noise statistics and noise mismatch issues. The proposed method was applied to predict the biological demand of oxygen, chemical demand of oxygen, suspended solids in the effluent of a real wastewater treatment plant, and obtain good monitoring performance. The obtained results demonstrate that the Elman-SR-UKF algorithm can produce more accurate prediction results and has a faster convergence than standard Elman neural networks, what is more, alleviating the problem of underprediction phenomena in the Elman network-based soft sensor. This may be due to a key strategy: The inherent statistical averaging of the Elman-SR-UKF algorithm can be less likely to get stuck in local minima.

To further restrict weight values within a certain and reasonable range and then to avoid large weight values, as well as improving the reliability of a soft sensor, weight constraint algorithm is further introduced to this study. The related results show that the weight constraint method successfully limits the parameter values and improves the reliability of a soft sensor. Nevertheless, the weight-limiting mechanism has an effect on the convergence speed and prediction performance of the model, which may be due to two aspects: The constrained weight values reach the saturation region or the weight constraint makes the model less sensitive to the error. Therefore, this method should be applied based on actual needs.

In the future, the Elman-SR-UKF can be further optimized with the node-decoupled SR-UKF to reduce the complexity and storage requirements for the Elman-SR-UKF to ensure that the quality-related variables can be used for online control. Additionally, this study extends the proposed works to other more challenging processes such as oxidation ditches (ODs) and sequencing bath reactors (SBRs).

**Author Contributions:** All authors contributed to the study conceptualization and design. Material preparation, data collection, and analysis were performed by Y.L. (Yiqi Liu), D.L. and Y.L. (Yan Li). The first draft of the manuscript was written by Y.L. (Yiqi Liu) and L.Y. The writing review and editing were performed by Y.L. (Yiqi Liu) and L.Y. The funding was provided by D.H. and Y.L. (Yiqi Liu). All authors have read and agreed the published version of the manuscript.

**Funding:** This research was funded by the National Natural Science Foundation of China (61873096, 62073145), Guangdong Basic and Applied Basic Research Foundation (2020A1515011057, 2021B1515420003), Guangdong Technology International Cooperation Project Application (2020A0505100024, 2021A0505060001). Fundamental Research Funds for the central Universities, SCUT (2020ZYGXZR034). Y.L. (Yiqi Liu) also thanks for the support of Horizon 2020 Framework Programme-Marie Skłodowska-Curie Individual Fellowships (891627).

**Data Availability Statement:** The data presented in this study are available in http://www.ics.uci.edu/~mlearn/MLRepository.html (accessed on 20 November 2021).

**Conflicts of Interest:** The authors declare no conflict of interest.

## Abbreviations

The following abbreviations are used in this manuscript:

| | |
|---|---|
| $BOD_5$ | biochemical oxygen demand for 5 days |
| COD | chemical oxygen demand |
| TN | total nitrogen |
| SVI | sludge volume index |
| ODs | oxidation ditches |
| SBRs | sequencing bath reactors |
| KF | Kalman filter |

| | |
|---|---|
| EKF | extended Kalman filter |
| UKF | unscented Kalman filter |
| SR-UKF | square-root unscented Kalman filter |
| NN | neural network |
| RNN | recursive neural network |
| RTRL | real-time recurrent learning |
| BPTT | back propagation through time |
| LM | Levenberg-Marquardt |
| SUT | scaled unscented transformation |
| RMSE | root mean square error |
| R | correlation coefficient |
| RMSSD | root mean of diagonal square sum |
| MR | multiple correlation coefficient |
| DQO-S | output chemical demand of oxygen |
| DBO-S | output biological demand of oxygen |
| SS-S | output suspended solids |
| Elman-BPTT | Elman network based on back propagation through time algorithm |
| Elman-GDM | Elman network based on momentum gradient descent algorithm |
| Elman-LM | Elman network based on Levenberg-Marquardt algorithm |
| RTRL-LM | real-time recurrent learning based on Levenberg-Marquardt algorithm |
| Elman-RTRL-LM | Elman network based on RTRL-LM |

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
