# Peer review of "Process Monitoring of Quality-Related Variables in Wastewater Treatment Using Kalman-Elman Neural Network-Based Soft-Sensor Modeling"

_water, doi:10.3390/w13243659_

Round 1

Reviewer 1 Report

Reviewer 1:

I recommend a major amendment at this level.

General comments:

The manuscript entitled Process Monitoring of Quality-related Variables in Wastewater Treatment Using Kalman-Elman Neural Network-based Soft-sensor Modelingwas reviewed. The work carried out in the manuscript is interesting and aimed at the uses of the proposed algorithm to update the weights of the Elman network and to improve the prediction performance.  However, the authors are suggested to undergo the following correction. The innovation and the importance of this work are not clearly highlighted in the abstract, introduction, and conclusions. Please work on this and prove to us why this work is valuable. Please remove all the multiple references. After that please check the manuscript thoroughly and eliminate all the lumps in the manuscript. This should be done by characterizing each reference individually. This can be done by mentioning 1 or 2 phrases per reference to show how it is different from the others and why it deserves mentioning. This comment is applied all over the paper. Better connect your research findings to previous works published in Water and in other top journals. Your manuscript is not well connected to Water literature. In your discussion section, please link your empirical results with a broader and deeper literature review. Too many abbreviations are used in the analysis and results. I recommend a nomenclature section for the abbreviations and variables used throughout the passage. Please carefully check, revise and improve the whole manuscript as there are few syntax/grammatical errors. The service of an expert in the use of English in scientific writings should be sought if necessary. Discussion of the results should provide useful insights. Highlights are necessary for this journal. It is highly recommended to provide a graphical abstract, as it will increase the visibility of the work and make the manuscript more appealing. The detailed comments can be seen below:

Detailed comments:

Title: Ok.

Abstract:

The abstract should state briefly the purpose of the research, the principal results, and major conclusions. An abstract is often presented separately from the article, so it must be able to stand alone. The abstract should include a sentence about your findings, discussions, and conclusions in your abstract and underscore the scientific value-added of your paper in your abstract. In the abstract, please add an indication of the achievements from your study that are relevant to the journal scope. Please be concise - maximum 1-2 lines. Data should be incorporated into the abstract.

Introduction:

Please improve the state of the art overview, to clearly show the progress beyond the state of the art. Please reason both the novelty and the relevance of your paper goals. The lack of proper justification creates the wrong impression that the authors are unaware of the recent developments. Please improve the aim of the introduction.

Please eliminate the use of redundant words. Eg. In this way, Recently, Respectively, therefore, currently, thus, hence, finally, to do this, first, in order, however, moreover, nowadays, today, consequently, in addition, additionally, furthermore. Please revise all similar cases, as removing these term(s) would not significantly affect the meaning of the sentence. This will keep the manuscript as CONCISE as possible. Please check ALL. Avoid beginning or ending a sentence with one or a few words, they are usually redundant. Kindly revise all.

Materials and Methods:

Please add in the beginning your scientific hypothesis. In the course of describing the performed actions, please provide reader guidance, sufficient for understanding why those actions have been performed. What is the preliminary materials and methods? It is better to change Materials and Methods. what is the reason the authors choose this method?

Results and Discussion:

This is the main problem. Why the authors never compare and validated their results with others?? In the section of the discussion, when discussed with results, the authors should improve the logic to make it readable. I recommend the authors spend some time analyzing the things in a better way by extending the discussion including a suitable mechanism for the variation of obtained results.

Conclusions:

The conclusion is pretty generic and fails to provide any improvement in the existing knowledge base. Limitations in the suggested approach should be discussed in the conclusions section. Please make sure your conclusions section underscores the scientific value-added of your paper, and/or the applicability of your findings/results. Highlight the novelty of your study.

References:

Reference needs more updating with current aspects of work being carried out. They need to strictly follow the journal's guidelines. Please extend and add more references especially in the results and discussion section.

Author Response

Thanks for reviewer 1's comments. The responses have been added in the following attached file.

Reviewer 2 Report

-Please provide the results of model fit calculations for the learning and test data,
- the algorithm in figure 1 should be clear and the reader should understand how to build the model and the connection of subsequent components, it is not clear how module A connects to B, or maybe they are independent modules, 
- it is difficult to assess the correspondence of the calculation results to the measurements; and the figures are missing, RMSE is an important measure of fit, but it is advisable to calculate also the average relative error, in figure 4 the data is missing

Author Response

Thanks for reviewer 2's comments. The responses have been added in the following attached file.

Round 2

Reviewer 1 Report

Reviewer Reports:

I reviewed the revised manuscript entitled" Process Monitoring of Quality-related Variables in Wastewater Treatment Using Kalman-Elman Neural Network-based Soft-sensor Modeling". The authors have carefully addressed and explained Most of the comments. There is only two more comments authors need to consider before possible publication:

Details Comments:

1- Data (please add numerical data) should be incorporated in the abstract.

2- Please compare more your works with others in the results and discussion section.

Author Response

Thanks for the reviewers’ comments. We have attached the revision notes accordingly.

Reviewer 2 Report

-Figure 4 is not clear, these are time series ???? is not clear, are series of arbitrary values ???
- in Figure 1, the authors separate the Elman network into two modules: A, B; I am looking for a reference in the manuscript to these separated modules and cannot find a reference in the text to module A and B

Author Response

(The authors gave the same response as above.)
